# Porcine sapovirus-induced RIPK1-dependent necroptosis is proviral in LLC-PK cells

**Muhammad Sharif**[1◉], **Yeong-Bin Baek**[1◉], **Thu Ha Nguyen**[1], **Mahmoud Soliman**[1,2]*, **Kyoung-Oh Cho**[1]*

**1** Laboratory of Veterinary Pathology, College of Veterinary Medicine, Chonnam National University, Gwangju, Republic of Korea, **2** Faculty of Veterinary Medicine, Department of Pathology & Clinical Pathology, Assiut University, Assiut, Egypt

◉ These authors contributed equally to this work.
* mahmoud.soliman@vet.au.edu.eg (MS); choko@chonnam.ac.kr (KOC)

**Data Availability Statement:** All relevant data are within the manuscript and its Supporting Information files.

**Funding:** K.-O.C. was supported by a grant (grant 2020R1A2B5B03002517 from the Basic Science

## Abstract

Sapoviruses belonging to the genus *Sapovirus* within the family *Caliciviridae* are commonly responsible for severe acute gastroenteritis in both humans and animals. Caliciviruses are known to induce intrinsic apoptosis in vitro and in vivo, however, calicivirus-induced necroptosis remains to be fully elucidated. Here, we demonstrate that infection of porcine kidney LLC-PK cells with porcine sapovirus (PSaV) Cowden strain as a representative of caliciviruses induces receptor-interacting protein kinase 1 (RIPK1)-dependent necroptosis and acts as proviral compared to the antiviral function of PSaV-induced apoptosis. Infection of LLC-PK cells with PSaV Cowden strain showed that the interaction of phosphorylated RIPK1 (pRIPK1) with RIPK3 (pRIPK3), mixed lineage kinase domain-like protein (pMLKL) increased in a time-dependent manner, indicating induction of PSaV-induced RIPK1-dependent necroptosis. Interfering of PSaV-infected cells with each necroptotic molecule (RIPK1, RIPK3, or MLKL) by treatment with each specific chemical inhibitor or knockdown with each specific siRNA significantly reduced replication of PSaV but increased apoptosis and cell viability, implying proviral action of PSaV-induced necroptosis. In contrast, treatment of PSaV-infected cells with pan-caspase inhibitor Z-VAD-FMK increased PSaV replication and necroptosis, indicating an antiviral action of PSaV-induced apoptosis. These results suggest that PSaV-induced RIPK1-dependent necroptosis and apoptosis–which have proviral and antiviral effects, respectively–counterbalanced each other in virus-infected cells. Our study contributes to understanding the nature of PSaV-induced necroptosis and apoptosis and will aid in developing efficient and affordable therapies against PSaV and other calicivirus infections.

## Introduction

Caliciviruses are small, non-enveloped, icosahedral viruses with a single-stranded, positive-sense RNA genome of 7 to 8 kb in size [1]. They have been divided into 11 genera [2]; 7 genera that infect mammals include *Norovirus*, *Sapovirus*, *Lagovirus*, *Vesivirus*, *Nebovirus* [1],

Research Program through the National Research Foundation of Korea, which is funded by the Ministry of Science, ICT and Future Planning, Republic of Korea.

**Competing interests:** The authors have declared that no competing interests exist.

*Recovirus* [3], and *Valovirus* [4]; 2 genera that infect birds are *Bavovirus* [5, 6] and *Nacovirus* [6–8]; and 2 genera that infect fish are *Salovirus* [9] and *Minovirus* [10]. Sapoviruses and noroviruses are important causative agents of viral gastroenteritis in humans and animals [1, 11]. The inability of sapoviruses to grow in cell culture has hindered our understanding of their biology [11, 12]. Porcine sapovirus (PSaV) Cowden strain is the only cultivable strain within the genus *Sapovirus*. It replicates efficiently in vitro in a porcine kidney cell line LLC-PK supplemented with porcine intestinal contents or bile acids, specifically glycochenodeoxycholic acid (GCDCA) [12, 13]. Therefore, PSaV Cowden strain is an appealing model to study the life cycles of sapoviruses and for use in developing vaccines or therapeutic interventions against human and animal sapoviruses.

The cell can die either by accidental cell death through severe insults of physical, chemical, and mechanical stimuli or regulated cell death (RCD) via dedicated molecular machinery [14]. The Nomenclature Committee on Cell Death has proposed an updated classification of fifteen cell death subroutines of RCD [14]. Four forms of RCD occur as a result of viral infections: apoptosis, necroptosis, pyroptosis, and ferroptosis [15–17]. Apoptosis is the most extensively studied RCD subroutine in the context of virus infection [15, 18]. Morphologically, apoptosis has characteristic features such as condensation of chromatin, fragmentation of DNA, exposure of phosphatidylserine to the plasma membrane, and the formation of apoptotic bodies [14, 19, 20]. Apoptosis is induced by the action of the executioner caspase, caspase 3, which is individually activated by the two initiator caspases, caspase 8 through the extrinsic pathway or caspase 9 through the intrinsic pathway, respectively [14, 18, 21]. Necroptosis is a recently discovered RCD subroutine [14]. It is morphologically different from apoptosis and is characterized by swelling of organelles and rupture of the plasma membrane to release intracellular contents [14, 15]. Induction of necroptosis requires the function of receptor-interacting protein kinase 1 (RIPK1). Once activated, RIPK1 interacts with RIPK3 through RIPK homotypic interaction motif (RHIM), leading to their autophosphorylation and formation of a complex known as the necrosome [22, 23]. Phosphorylated RIPK3 (pRIPK3) recruits and phosphorylates the mixed lineage kinase-like protein (MLKL), which translocates to and ruptures the plasma membrane [24–26]. This process is known as RIPK1-dependent necroptosis [14, 27]. Although both RIPK1 and RIPK3 are necessary for induction of necroptosis, RIPK3 can alone promote RIPK3-dependent necroptosis, in which RIPK3 forms complexes with DNA-dependent activator of interferon (IFN) regulatory factor (IRF) and the adaptor molecule TIR-domain-containing adaptor inducing IFN-b (an adaptor protein downstream of Toll-like receptor 3 [TLR3]) [14, 27, 28].

Viral infections activate and assemble cytosolic inflammasome, which mediates the activation of caspase-1. Activated caspase-1 leads to the maturation of pro-interleukin 1β (IL-1β) and pro-interleukin-18 (IL-18) and cleaves a gasdermin-D [14, 29]. The N-terminal region of cleaved gasdermin-D makes oligomer pores by inserting into the cytoplasm membrane to release mature IL-1β and IL-18 into the extracellular space. Fore formation by gasdermin-D induces osmotic imbalances and membrane ruptures, leading to pyroptosis, which is morphologically similar to necroptosis [14, 29]. Ferroptosis, a mechanism of RCD resulting from iron-mediated oxidative perturbations of the intracellular microenvironment, has been reported in Newcastle disease virus-infected tumor cells via activation of the p53-SLC7A11-GPX4 pathway and has been speculated to occur in many other RNA and DNA viruses because of alteration of intra- and/or extracellular iron levels or its associated molecules during their life cycle [17, 29, 30].

Since rabbit hemorrhage disease virus (RHDV) within the genus *Largovirus* has been first reported to induce hepatocyte apoptosis in RHVD-challenged rabbits, caliciviruses in the genera *Norovirus*, *Vesivirus*, and *Lagovirus* have been known to cause intrinsic apoptosis *in vitro* and *in vivo* [31–38]. Calicivirus-induced apoptotic cells are characterized morphologically by

chromatin condensation, DNA fragmentation, plasma membrane blebbing, and cell shrinkage and evidenced molecularly by sequential activation of initiator caspase 9 and effector caspase 3 after translocation and formation of Bax channel and leakage of cytochrome C and other apoptosis-associated proteins from the permeabilized mitochondria into the cytoplasm [31, 33, 35, 37–42]. In contrast, to our knowledge, there are no reports of necroptosis induced by any caliciviruses. This prompts us to investigate whether the PSaV Cowden strain as a representative of caliciviruses induces necroptosis in virus-infected LLC-PK cells and its role in PSaV replication as proviral or antiviral factors. Here, we demonstrated that PSaV Cowden strain induced RIPK1-dependent necroptosis as a proviral factor. In addition, apoptosis induced by the PSaV Cowden strain acts antiviral function. Our study provides novel insights into the mechanisms of calicivirus-induced necroptosis and highlights their importance for developing therapeutic drugs.

## Materials and methods

### Cells and virus

LLC-PK cells obtained from the American Type Culture Collection (ATCC, Manassas, VA, USA) were grown in Minimum essential medium (MEM) (Welgene, Daegu, South Korea) supplemented with 10% fetal bovine serum (FBS), 100 U/mL penicillin, and 100 μg/mL streptomycin at 37˚C in a 5% $CO_2$ atmosphere.

PSaV Cowden strain was recovered from a full-length infectious clone pCV4A [13] and propagated in LLC-PK cells supplemented with 200 μM GCDCA (Sigma Aldrich, St. Louis, MO, USA), 2.5% FBS, 100 U/mL penicillin, and 100 μg/mL streptomycin [43]. The virus was then concentrated by ultracentrifugation. The viral titer was determined by median tissue culture infective dose ($TCID_{50}$) assay as described previously [44] and calculated by the method of Reed and Muench [45].

### Reagents and antibodies

Necrostatin-1 (Nec-1), GSK'872, necrosulfonamide (NSA), and Z-VAD-FMK were purchased from Calbiochem (Darmstad, Germany) and dissolved in dimethyl sulfoxide (DMSO). Mouse monoclonal antibody (Mab) against RIPK1, rabbit Mab against MLKL, and rabbit polyclonal antibodies against RIPK3 and pRIPK3 were obtained from Abcam (Cambridge, UK), and rabbit Mabs against pRIPK1, pMLKL, and caspase 3 were purchased from Cell Signaling (Beverly, MA, USA). Mouse Mab against pMLKL was purchased from Novus Biologicals (Littleton, CO, USA). Mouse anti-glyceraldehyde 3-phosphate dehydrogenase (GAPDH, 0411) Mab was purchased from Santa Cruz (Dallas, TX, USA). Hyperimmune rabbit serum raised against PSaV VPg was produced by immunization of a New Zealand White rabbit [44, 46]. Secondary antibodies included horseradish peroxidase (HRP)-conjugated goat anti-rabbit immunoglobulin G (IgG) antibody (Cell Signaling), HRP-conjugated goat anti-mouse IgG antibody (Ab Frontier, Seoul, South Korea), Alexa Fluor (AF) 594-conjugated donkey anti-rabbit IgG and AF488-conjugated goat anti-mouse IgG were obtained from Life Technologies (Eugene, OR, USA). AF488-labeled phalloidin and slowFade Gold antifade reagent with 4',5-diamidino-2-phenylindole (DAPI) was purchased from Molecular Probes (Eugene, OR, USA). Protein A/G agarose beads were purchased from Santa Cruz.

### Cytotoxicity and cell viability assays

The 3-(4,5-dimethylthiazol-2-yl)-2,5-diphenyl tetrazolium bromide (MTT) assay was used to determine the cytotoxic effects of the chemicals and their solvents as well as their effect on cell

viability following PSaV Cowden strain infection, as previously described [46, 47]. Briefly, monolayers of LLC-PK cells grown in 96-well plates were incubated for 24 h with a medium containing different concentrations of each chemical inhibitor. For the cell viability assay, cells grown in 96-well plates were infected with PSaV Cowden strain at a multiplicity of infection (MOI) of 1 $TCID_{50}$, treated with or without each chemical inhibitor in a dose-dependent manner, and incubated for the indicated time points. After removing the medium, 200 μL of MTT solution was added to each well and incubated for 4 h at 37˚C in a $CO_2$ incubator. Then, each well was incubated with 150 μL of DMSO and incubated for 10 min at 20˚C. Absorbance was measured using an enzyme-linked immunosorbent assay (ELISA) reader at an optical density (OD) of 570 nm. The percent cell viability was calculated using the following formula: $[(OD_{sample}—OD_{blank})/(OD_{control}—OD_{blank})] \times 100$. All chemicals were used at concentrations that did not alter cell viability. In each experiment, the chemicals were freshly diluted with media to the desired concentration before adding to the cell monolayers.

## Treatment of LLC-PK cells with chemical inhibitors

LLC-PK cells grown in 6- or 96-well plates were washed twice with phosphate-buffered saline (PBS, pH 7.4). Cells were either infected with the PSaV Cowden strain or left mock-infected and treated with the inhibitory chemicals for 1 h at 37˚C or vehicle at the following concentrations: RIPK1 (Nec-1; 20 μM), RIPK3 (GSK'872; 10 μM), MLKL (NSA; 10 μM), and pan-caspase (Z-VAD-FMK; 20 μM) were added to the fresh cell culture medium for the indicated time periods as described elsewhere [47, 48]. Cells were then used to measure protein expression by western blotting, viral titer by $TCID_{50}$ assay, cell viability by MTT assay, and the percentage of apoptotic and necroptotic cells by flow cytometry assay.

## Transfections of siRNAs

LLC-PK cells were cultured in 6-well culture plates at 70–80% confluence and transfected with scrambled siRNA or siRNAs against RIPK1, RIPK3, or MLKL [46, 47] using Lipofectamine 3000 reagent (Invitrogen) following the manufacturer's instructions. The knockdown efficiency of each siRNA was optimized by a second transfection 24 h after the first transfection. The cells were then infected or not with the PSaV Cowden strain (MOI = 1 $TCID_{50}$). After 1 h, the viral inocula were removed and the cells were maintained in MEM with 200 μM GCDCA. The cells were harvested 24 h after infection, and viral protein expression and viral titer were measured by western blot analysis and $TCID_{50}$, as described below.

## Western blot analysis

Western blot analysis was performed to determine viral and host target proteins as described elsewhere [47]. In brief, confluent LLC-PK monolayers grown in 6-well plates were infected with or without PSaV Cowden strain, treated with or without various inhibitors, or transfected with scrambled siRNA or siRNAs against target proteins as described above. The cells were then washed three times with cold PBS (pH 7.4) and lysed using cell extraction buffer (10 mM Tris/HCl pH 7.4, 100 mM NaCl, 1 mM EDTA, 1 mM EGTA, 1 mM NaF, 20 mM $Na_2P_2O_7$, 2 mM $Na_3VO_4$, 1% Triton X-100, 10% glycerol, 0.1% SDS, and 0.5% deoxycholate (Invitrogen)) supplemented with protease and phosphatase inhibitors (Roche, Basel, Switzerland) for 30 min on ice. Cell lysates were centrifuged at $12,000 \times g$ for 10 min at 4˚C. The supernatants were analyzed for total protein content by using a bicinchoninic acid protein assay kit (ThermoFisher Scientific, Waltham, MA, USA). Denatured cell lysates were resolved by sodium dodecyl sulfate-polyacrylamide gel (SDS-PAGE) and transferred to nitrocellulose membranes (GE Healthcare Life Sciences, Piscataway, NJ, USA). The membranes were blocked for 1 h at

room temperature with Tris-buffered saline containing 5% skim milk before they were incubated overnight at 4˚C with the indicated primary antibodies. HRP-labeled secondary antibodies were added for 1 h at 20˚C and the immunoreactive bands were detected by an enhanced chemiluminescence reaction kit (Dogen, Seoul, South Korea) using the Davinch-K Imaging System (Youngwha Scientific Co., Ltd, Seoul, South Korea).

## Terminal deoxynucleotidyl transferase dUTP nick end labeling (TUNEL) assay

The percentage of apoptotic cells was determined by TUNEL assay [47, 49] using an *in situ* Cell death detection kit (Roche) according to the manufacturer's protocol. Briefly, LLC-PK cells were infected with or without PSaV Cowden strain (MOI = 1 TCID$_{50}$), harvested at different time points, fixed, and permeabilized as described above. The cells were then incubated with 50 μL of TUNEL reaction mixture (1:9 ratio of enzyme solution to label solution) for 60 min at 37˚C in the dark. Negative controls were made by omitting the terminal deoxynucleotidyl enzyme and using the same volume of the label solution. The cells were analyzed using a confocal microscope or an Attune$^{TM}$ NxT flow cytometer.

## Immunofluorescence (IF) assay

LLC-PK grown in 8-well chamber slides were treated with GCDCA or infected with or without PSaV Cowden strain (MOI = 1 TCID$_{50}$) for the indicated time points. Virus inocula were removed, and the cells were washed twice with PBS (pH 7.4), fixed with 4% paraformaldehyde for 15 min, permeabilized by the addition of 0.2% Triton X-100 for 5 min at 20˚C, and washed again with PBS. The cells were blocked with 5% bovine serum albumin for 1 h and then incubated with the TUNEL reaction mixture containing FITC-conjugated dUTP for apoptotic cells or mouse Mab against pMLKL for necroptotic cells. For co-staining with viral antigen, the cells were incubated with rabbit anti-PSaV VPg hyperimmune serum with the apoptotic TUNEL reaction mixture or necroptotic marker. After washing twice with PBS (pH 7.4), the cells were incubated with the secondary antibodies for 1 h at 20˚C and mounted with Slow-Fade Gold antifade reagent containing 1× DAPI solution (Molecular Probes) for nuclear staining. The infected cells were observed using an LSM 510 confocal microscope and analyzed using the LSM software (Carl Zeiss). The number of apoptotic and necroptotic cells was quantified by counting 100 cells from each sample and expressed as a percentage of the total population [47].

## Flow cytometry assay

A flow cytometry assay was used to determine the percentage of apoptotic and necroptotic cells as described elsewhere [50, 51]. Briefly, confluent LLC-PK cells grown on 6-well plates were treated with or without chemicals or infected with or without PSaV Cowden strain (MOI = 1 TCID$_{50}$), then washed twice with PBS and detached using Accutase$^{TM}$ Cell Detachment Solution (BD Biosciences, San Jose, CA, USA) at 37˚C for 5 min. Cells were collected and centrifuged at 500×g for 5 min and washed with ice-cold PBS. The cells were then divided into two groups for whole-cell analysis of apoptosis or necroptosis. Both groups of cells were fixed with ice-cold 4% paraformaldehyde for 15 min on ice, permeabilized with 0.2% Triton X-100 for 5 min at 20˚C, and then washed twice with PBS. For apoptosis analysis, the cells were incubated with 50 μL of TUNEL reaction mixture containing FITC-conjugated dUTP (In situ Cell death detection kit, Roche, Mannheim, Germany) for 60 min at 37˚C in the dark. For necroptosis analysis, the cells were incubated with rabbit pMLKL Mab (1:200 dilution) at 37˚C

for 1 h followed by incubation with AF488-conjugated donkey anti-rabbit IgG antibody (1:200 dilution) at 37˚C for 1 h.

To label the viral antigen, the cells were incubated with rabbit anti-PSaV VPg hyperimmune serum (1:200 dilution) at 37˚C for 1 h. The cells were then washed with PBS and incubated with AF488-conjugated donkey anti-rabbit IgG antibody (1:200 dilution) at 37˚C for 1 h. The stained cells were washed and analyzed directly by using an Attune[TM] NxT flow cytometer (Thermo Scientific) and the data from each sample were digitized using AttuneTM NxT software v3.1.2.

## Immunoprecipitation assay

Immunoprecipitation of the target protein was performed as previously described [47]. Briefly, LLC-PK cells grown in 6-well plates were infected with or without PSaV Cowden strain (MOI = 1 $TCID_{50}$), and then incubated for the indicated time points at 37˚C. The cells were lysed as described above and cell lysates were then precleared by incubation with protein A/G agarose beads (Santa Cruz) for 30 min at 4˚C. The precleared cell lysates were incubated with anti-pRIPK1 antibody overnight at 4˚C and the immune complexes were then captured by incubation with protein A/G agarose beads for 1 h at 4˚C. The immunoprecipitated proteins were then evaluated by western blot analysis as described above.

## Virus titration using $TCID_{50}$ assay

Virus titer was determined using $TCID_{50}$ assay as previously described [44, 46]. LLC-PK cells grown in 6-well plates were infected with or without PSaV Cowden strain (MOI = 1 $TCID_{50}$) and treated with or without chemical inhibitors or transfected with scrambled siRNA or siR-NAs against the target proteins. After three repeated freeze-thaw cycles, ten-fold serial dilutions of clarified virus supernatants were prepared in MEM supplemented with 200 μM GCDCA, 200 μL were inoculated to monolayers of LLC-PK cells grown in 96-well plates and incubated at 37˚C in a 5% $CO_2$ incubator. Virus titers were determined at 6 days post-infection and expressed as $TCID_{50}$/mL values by the method of Reed and Muench [45].

## Statistical analysis

All statistical analyses were performed on duplicate or triplicate experiments using one-way or two-way ANOVA and the GraphPad Prism software version 8.4.2 (GraphPad Software Inc., La Jolla, CA, USA). *P*-values of less than 0.05 were considered significant.

# Results

## PSaV infection induces both necroptosis and apoptosis in LLC-PK cells

Supplementation of bile acids such as GCDCA into the culture medium of PSaV Cowden strain-inoculated LLC-PK cells is essential to growing it [13]. To rule out a bias possibly due to cell death by the supplementation of GCDCA into the medium, we first performed an MTT assay to check cell viability of LLC-PK cells either infected with PSaV Cowden strain (MOI = 1 $TCID_{50}$) in the presence of 200 μM GCDCA or incubated just with medium containing 200 μM GCDCA. The PSaV-infected LLC-PK cells in the presence of GCDCA showed a significant decrease in cell viability at 18–36 h post-infection (hpi), whereas mock-infected cells treated with GCDCA at 200 μM concentration remained viable up to 36 h following treatment (Fig 1A). These results indicated that PSaV replication induced cell death in the presence of GCDCA but the treatment of cells with GCDCA alone had no influence on cell death. Next, we examined the nature of cell death during PSaV infection. LLC-PK cells mock-infected or infected with PSaV in the presence of 200 μM GCDCA were harvested, lysed, and subjected to western blotting. Interestingly, the

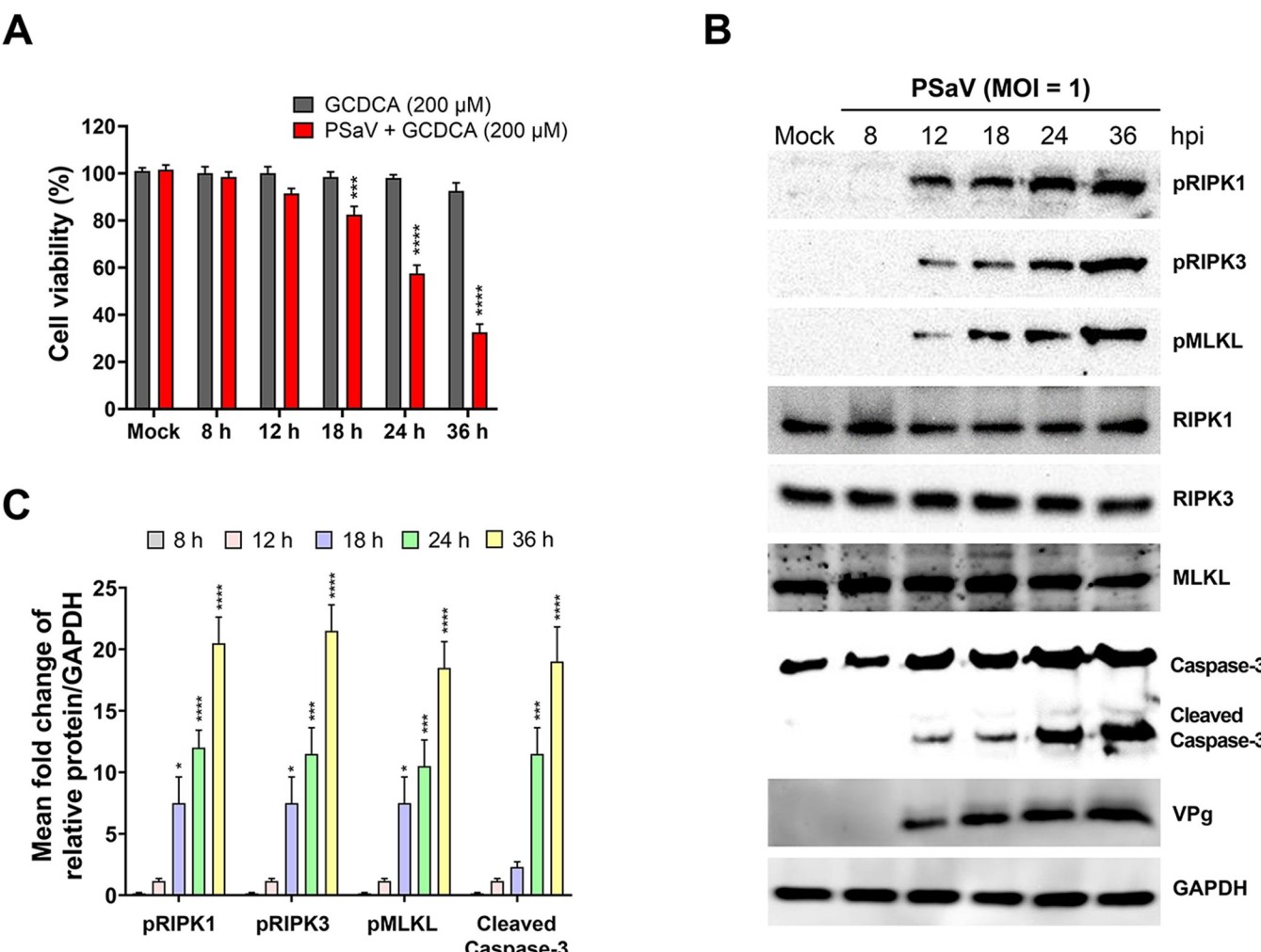

**Fig 1. PSaV induced activation of necroptosis and apoptosis key molecules in LLC-PK cells.** (A) LLC-PK cells were either treated with 200 μM glycochenodeoxycholic acid (GCDCA) alone or treated with 200 μM GCDCA after infection with PSaV Cowden strain at a MOI of 1 TCID$_{50}$. As a negative control, LLC-PK cells were mock-treated and mock-infected. The cell viability was measured by MTT assay (using ELISA reader at OD$_{570}$) at the indicated time points, and the results are expressed as the mean percentages of viable cells for three independent experiments. (B) LLC-PK cells were infected with PSaV Cowden strain (MOI = 1 TCID$_{50}$) in the presence of 200 μM GCDCA and the cell lysates were collected at the indicated time points and then subjected to western blot analysis using the indicated antibodies. GAPDH was used as a loading control. Results are representative of two independent experiments. (C) The relative expression of pRIPK1, pRIPK3, pMLKL, and cleaved caspase-3 in virus-infected cells (in panel B) was determined via densitometric analysis. Data in panels A and C represent the means ± standard error from two independent experiments. Differences were evaluated using the two-way ANOVA. $^*p < 0.05$; $^{***}p < 0.001$; $^{****}p < 0.0001$.

main signal molecules in the RIPK1-dependent necroptosis pathway RIPK1, RIPK3, and MLKL were phosphorylated and increased in response to PSaV infection in a time-dependent manner from 12 to 36 hpi, compared to mock-infected cells (Fig 1B and 1C). Cleaved caspase 3, the crucial mediator of the extrinsic and intrinsic apoptotic pathways, was detected from 12 to 36 hpi (Fig 1B and 1C), which was not detected in mock-infected cells. These results suggested that PSaV could induce both necroptosis and apoptosis in virus-infected LLC-PK cells.

## PSaV infection directly induces necroptosis and apoptosis in LLC-PK cells

To investigate the dynamic of necroptotic and apoptotic cells during PSaV infection, they were first either immune-stained with the anti-pMLKL antibody or labeled with the TUNEL

method and then quantified using flow cytometry. Results showed a time-dependent increase in the number of necroptotic and apoptotic cells in response to PSaV infection (Fig 2A–2C). We next determined whether both cell death modalities are due to PSaV replication. Of note, the number of cells showing dual positive reactions for PSaV VPg antigen with necroptotic marker pMLKL or apoptotic marker TUNEL increased in a time-dependent manner (Fig 2D–2F). We further performed IF assay to visualize the apoptotic and necroptotic cells and determine their dynamic of during PSaV replication in LLC-PK cells infected with PSaV in the presence of GCDCA using confocal microscopy. As expected, neither apoptotic nor necroptotic cells were detected in GCDCA-treated cells at the indicated times (Fig 3A and 3B). However, the number of cells positive for both PSaV VPg and apoptotic marker (Fig 3C and 3E) or necroptotic marker (Fig 3D and 3F) gradually increased in a time-dependent manner. Interestingly, pMLKL, the major executioner factor for rupturing plasma membrane in the necroptotic pathway [26] increased in the expression levels and then translocated to the surface of inner plasma membrane in the PSaV-infected cells (Fig 3C). Taken together, our findings demonstrate that PSaV could directly induce both necroptosis and apoptosis.

## PSaV-induced necroptosis via the RIPK1/RIPK3/MLKL pathway is proviral

To investigate whether the key molecules involved in necroptosis interact during PSaV infection, confluent LLC-PK cells were either mock-infected or infected with PSaV Cowden strain (MOI = 1 TCID$_{50}$), and cell lysates were immunoprecipitated with an anti-RIPK1 antibody at the indicated times. The interaction of pRIPK1 with pRIPK3 and pMLKL was evaluated through western blot analysis. Results showed that the pRIPK1 antibody immunoprecipitated pRIPK3 and pMLKL in virus-infected cells (Fig 4A) and the interaction of pRIPK1 with pRIPK3 or pMLKL quantified by densitometric analysis was gradually increased in a time-dependent manner (Fig 4B and 4C). These findings demonstrated that PSaV-induced necroptosis is mediated by the RIPK1-dependent pathway.

Necroptosis and apoptosis are generally considered to be antiviral mechanisms, but they may also benefit viruses in certain situations by promoting viral release and spread [52–55]. We examined the role of necroptosis and apoptosis in PSaV replication using chemical inhibitors including RIPK1 inhibitor Nec-1, RIPK3 inhibitor GSK'872, MLKL inhibitor NSA, and pan-caspase inhibitor Z-VAD-FMK. We first measured their cytotoxicity using MTT assay and found that the optimal concentrations of Nec-1, GSK'872, NSA, and Z-VAD-FMK were 30, 10, 10, and 20 µM, respectively (Fig 5). Treatment of PSaV-infected LLC-PK cells with Nec-1, GSK'872, NSA, and Z-VAD-FMK significantly reduced the expression of pRIPK1, pRIPK3, pMLKL, and cleaved caspase-3, respectively (Fig 6A–6D). Interestingly, inhibition of necroptosis by the treatment with each chemical necroptosis inhibitor suppressed the expression of PSaV VPg protein (Fig 6E and 6F) and PSaV titer (Fig 6G). However, treatment with chemical apoptosis inhibitor increased viral protein expression and titer (Fig 6E–6G). We further confirm the above results by examining the effect of siRNAs against RIPK1, RIPK3, and MLKL on PSaV replication. Western blotting results revealed that these siRNAs effectively silenced RIPK1, RIPK3, and MLKL, respectively, in either mock-infected LLC-PK cells (Fig 7A–7C) or PSaV-infected cells (Fig 7D–7F). Knockdown of them also decreased the viral protein expression and viral titer (Fig 7G–7I). Taken together, our data suggest that PSaV-induced RIPK1-dependent necroptosis has a proviral nature, in contrast to antiviral apoptosis.

## Counterbalancing of PSaV-induced necroptosis and apoptosis

A shift in cell death modality from apoptosis to necroptosis and vice versa has been observed in response to human immunodeficiency virus type-1 (HIV-1), Theiler's murine

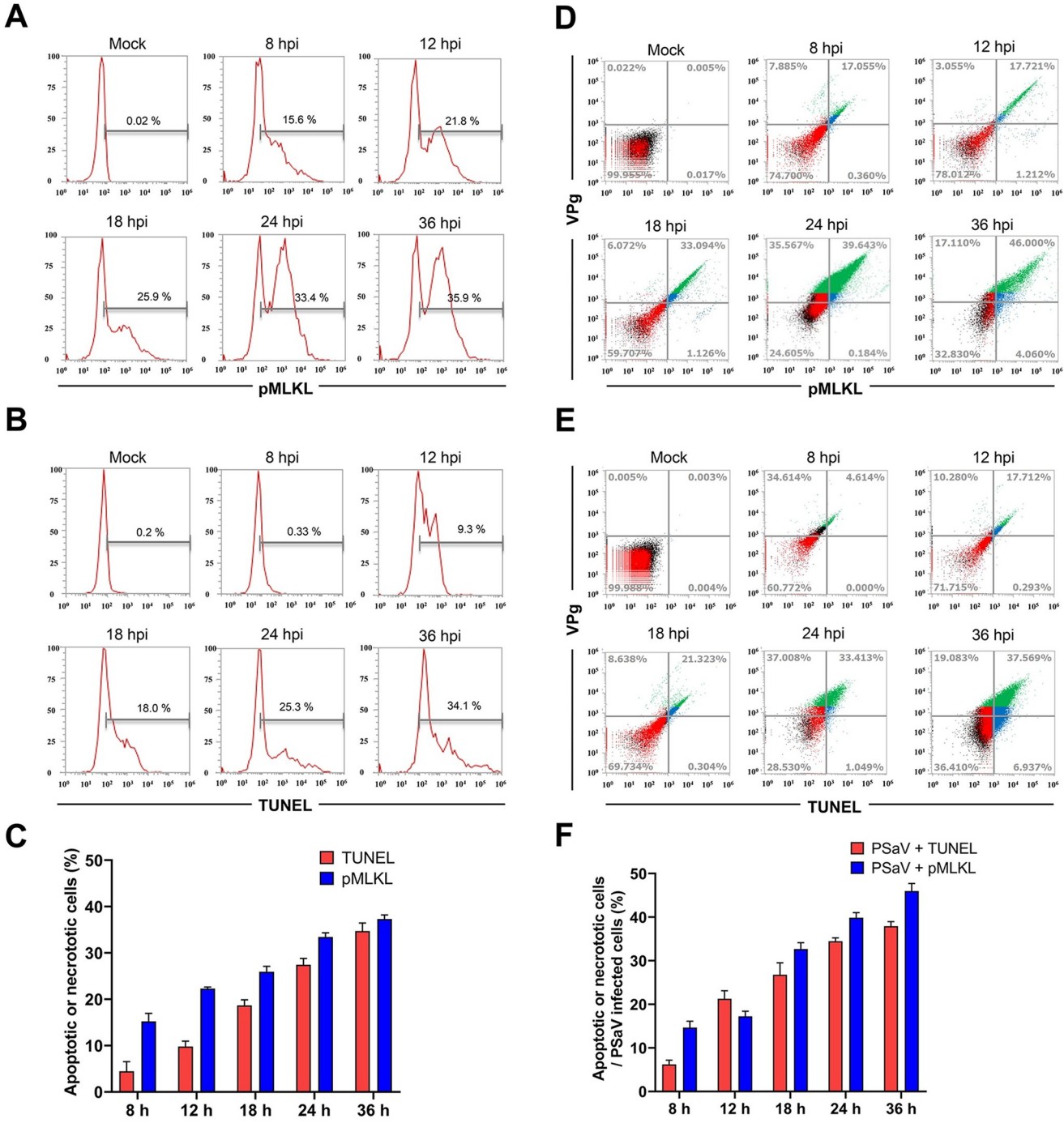

**Fig 2. Dynamics of PSaV-induced necroptosis and apoptosis.** (A, B) LLC-PK cells either infected with PSaV Cowden strain (MOI = 1 TCID$_{50}$) or left mock-infected in the presence of 200 μM GCDCA were harvested at the indicated time points and analyzed using flow cytometry to quantify the number of necroptotic cells using pMLKL antibody (A) or apoptotic cells using TUNEL staining (B). (C) The graphical representation of the necroptotic and apoptotic cell populations in panels A and B. (D, E) The LLC-PK cells harvested in the above condition were co-labeled for the PSaV VPg antigen (using VPg antibody) and either TUNEL or pMLKL antibody and analyzed by flow cytometry to detect PSaV-positive necroptotic (D) or apoptotic (E) cells. (F) The graphical representation of the PSaV VPg-positive necroptotic and PSaV VPg-positive apoptotic cell populations in panels D and E. Data are shown as the percentage of necrotic or apoptotic cells in PSaV infected cells. The data in panels C and F represent the means ± standard error from two independent experiments.

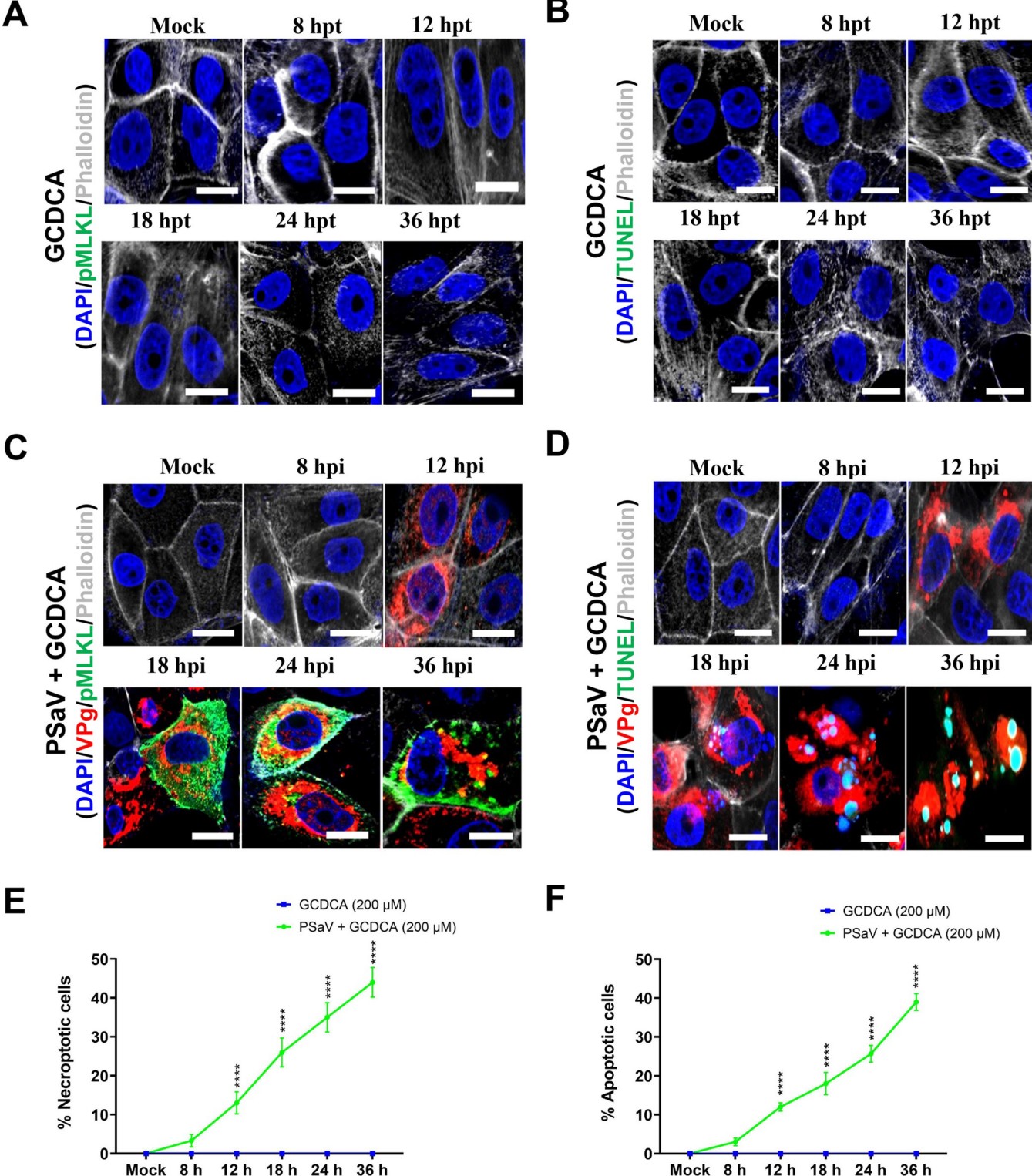

**Fig 3. PSaV-induced apoptosis and necroptosis examined by confocal microscopy.** (A, B) LLC-PK cells were treated with 200 μM GCDCA for the indicated time points and processed for IF assay to detect necroptotic cells using an anti-pMLKL antibody (A) or for TUNEL assay to detect apoptotic cells (B). (C, D) LLC-PK cells mock-infected or infected with PSaV Cowden strain (MOI = 1 TCID$_{50}$) in the presence of 200 μM GCDCA and incubated for the indicated time points were processed for IF assay to detect the dual positive necroptotic cells with PSaV VPg antigen and pMLKL necroptotic marker (C) or for IF and TUNEL assays to detect the dual positive apoptotic cells with PSaV VPg antigen and TUNEL marker (D). The nuclei were stained with DAPI. Actin

cytoskeleton was stained with phalloidin. (E, F) The percentage of PSaV VPg antigen-positive apoptotic (E) or PSaV antigen-positive necroptotic (F) LLC-PK cells either infected or left mock-infected in the presence of 200 μM GCDCA were evaluated by counting the cells stained by TUNEL or pMLKL, respectively, as mentioned in the materials and methods. The data in panels E and F represent the means ± standard error from two independent experiments. Differences were evaluated using two-way ANOVA. ****$p < 0.0001$.

encephalomyelitis virus (TMEV), and rotavirus infections in the presence of chemicals against the necroptosis or apoptosis [47, 53, 55, 56]. As shown in Figs 6 and 7, a decrease in PSaV protein level and titer following the downregulation of necroptosis molecules (RIPK1, RIPK3, or MLKL) likely resulted from cell death protection, induction of apoptosis, or both. In order to examine this hypothesis, we performed an MTT assay to check cell viability and flow cytometry to determine the proportion of necroptotic and apoptotic cell death in cells infected with PSaV and treated with or without each inhibitor. Inhibition of necroptosis or apoptosis in PSaV-infected cells by individual treatment with each inhibitor did not prevent cell death by PSaV but lessened its impact (Fig 8A–8D) and changed the proportion of cells that passed through apoptosis or necroptosis (Fig 9A-9D). Taken together, both necroptosis and apoptosis induced by PSaV function oppositely as proviral and antiviral, respectively, and counterbalance each other in infected cells.

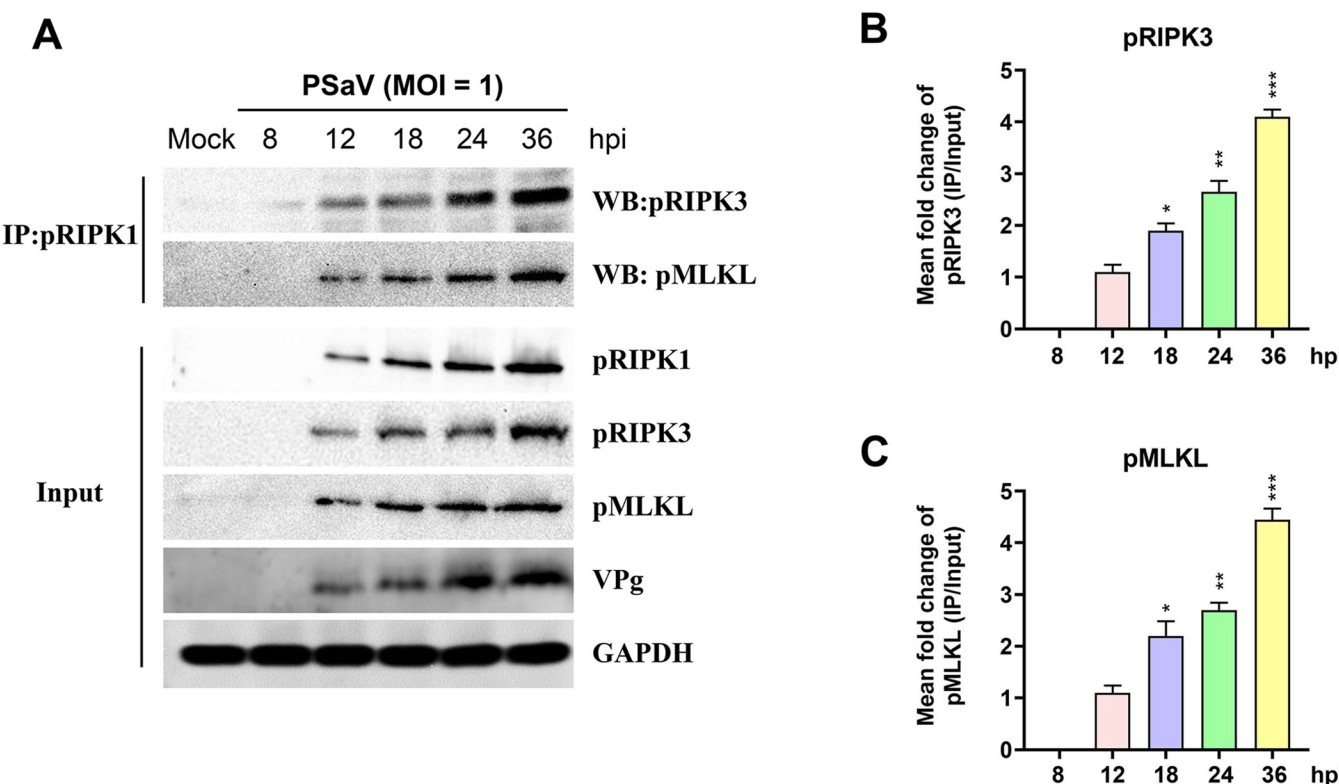

**Fig 4. PSaV-induced necrosome formation (RIPK1/RIPK3/MLKL complex).** (A) Lysates of LLC-PK cells infected with PSaV Cowden strain (MOI = 1 TCID$_{50}$) or left mock-infected in the presence of 200 μM GCDCA were immunoprecipitated using an anti-pRIPK1 antibody at indicated time points. Immunoprecipitated pRIPK3 and pMLKL with pRIPK1 were determined using western blot analysis with relevant antibodies. Representative images of different gels from two independent experiments are presented. (B, C) The relative level of immunoprecipitated pRIPK3 or pMLKL with pRIPK1 was estimated by densitometric analysis. Data in panels B and C are presented as means ± standard error from two independent experiments. Differences were evaluated using the one-way ANOVA. *$p < 0.05$; **$p < 0.01$; ***$p < 0.001$.

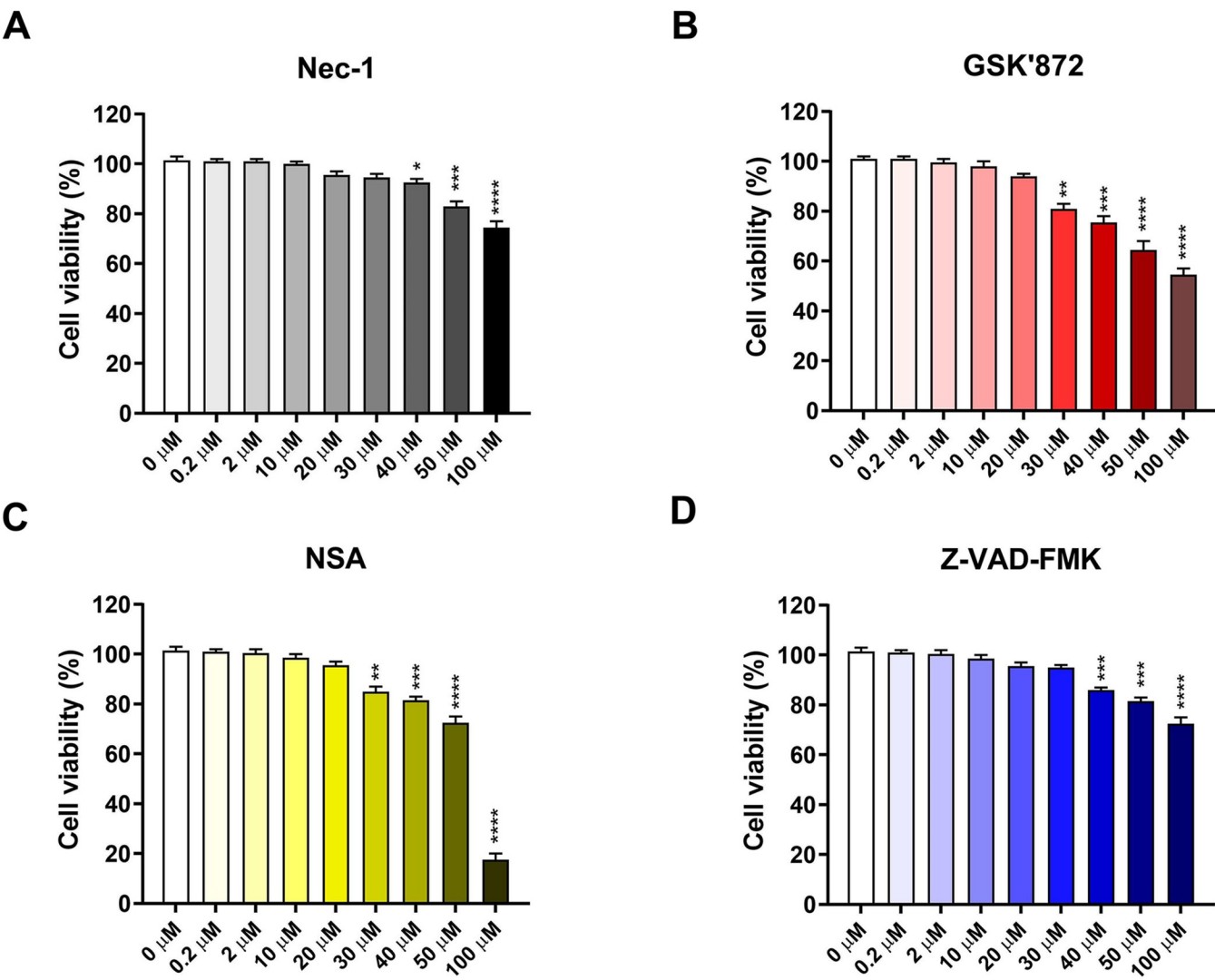

**Fig 5. Determination of the cytotoxicity of each necroptosis and apoptosis chemical by the MTT assay.** (A-D) LLC-PK cells grown in 96-well plates in triplicate were treated with various concentrations of the RIPK1 inhibitor Nec-1, RIPK3 inhibitor GSK'872, MLKL inhibitor NSA, and pan-caspase inhibitor Z-VAD-FMK or left mock-treated and incubated for 24 h at 37°C. Cell viability was measured at $OD_{570}$ using an ELISA reader. The results represent the means ± standard error from two independent experiments. Differences were evaluated using one-way ANOVA. $^*p < 0.05$; $^{**}p < 0.01$; $^{***}p < 0.001$; $^{****}p < 0.0001$.

## Discussion

The morphological and molecular features of intrinsic apoptotic cell death caused by calici-viruses in the genera *Norovirus*, *Vesivirus*, and *Lagovirus* have been well established [31–42]. However, calicivirus-induced necroptosis has not been studied so far. In the present study, interestingly, time-dependent gradual upregulation of pRIPK1/pRIPK3/pMLKL, major activated RIPK1-dependent necroptosis molecules, was determined by western blot analysis during PSaV replication in the LLC-PK cells. Moreover, the increase in the number of PSaV-infected necroptotic cell death was confirmed by flow cytometry and immunofluorescence assays. These data suggested that PSaV infection could directly induce RIPK1-dependent necroptosis. Of note, RIPK1-dependent necroptosis is known to occur after complex formation with pRIPK1, pRIPK3, and pMLKL [22–26]. Our immunoprecipitation assay confirmed a

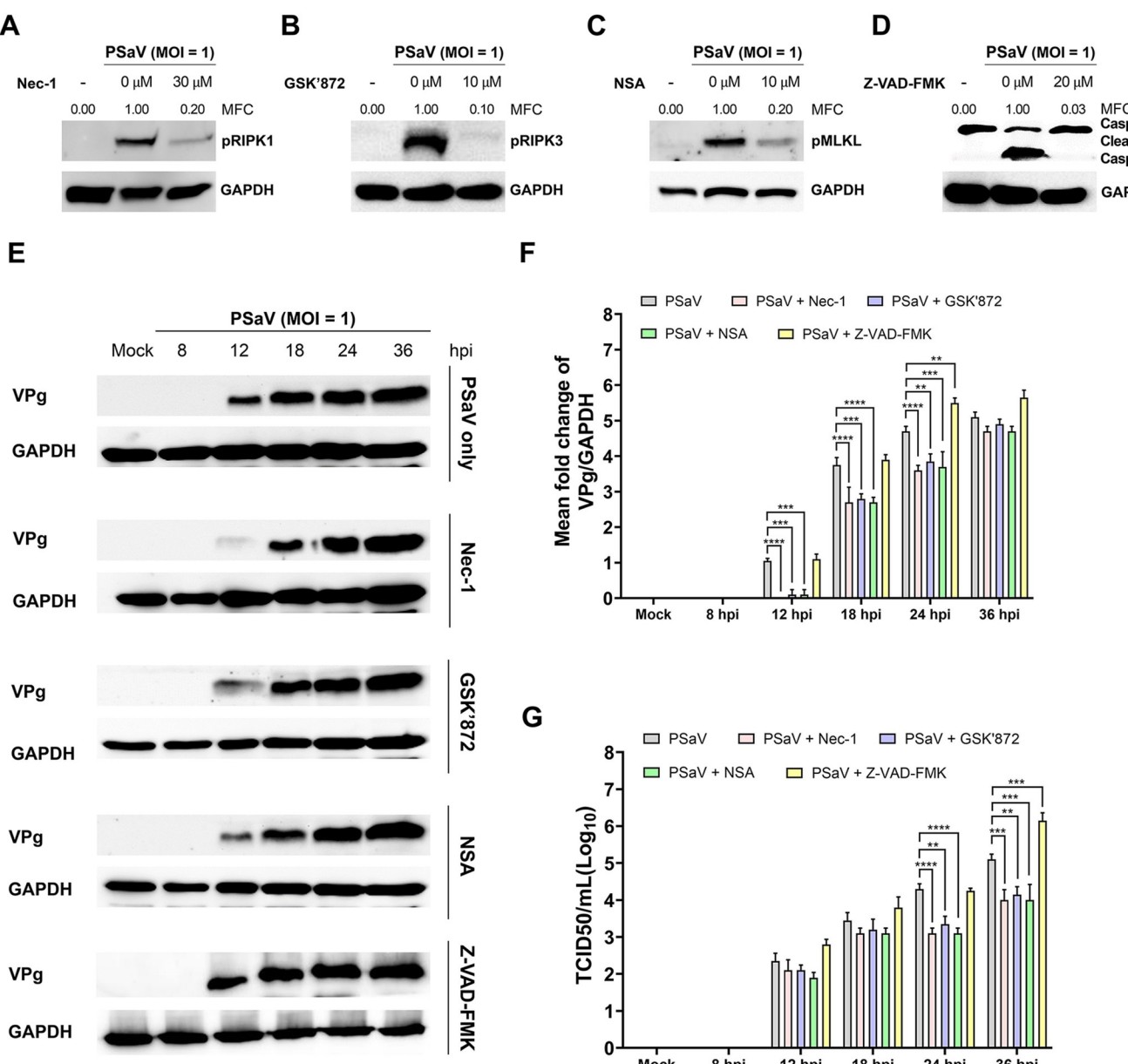

**Fig 6. Blocking the necrosome formation or apoptosis in LLC-PK cells inhibits or activates PSaV replication, respectively.** LLC-PK cells were infected with PSaV Cowden strain (MOI = 1 TCID$_{50}$) for 1 h and then incubated with a maintenance medium containing 200 μM GCDCA with 30 μM Nec-1 (RIPK1 inhibitor), 10 μM GSK'872 (RIPK3 inhibitor), 10 μM NSA (MLKL inhibitor), or 20 μM Z-VAD-FMK (pan-caspase inhibitor). (A-D) The cell lysates were harvested at 24 hpi and the expression levels of pRIPK1, pRIPK3, pMLKL, and cleaved caspase 3 were evaluated by western blotting. (E) The cell lysates at the indicated time points were subjected to western blotting to determine PSaV VPg protein. GAPDH was used as a loading control. (F) The relative expression of PSaV VPg in chemical-treated and virus-infected cells compared to that in mock-treated and virus-infected cells (in panel E) was determined via densitometric analysis. (G) The viral titer was determined using the TCID$_{50}$ assay and compared with that in the mock-treated and virus-infected cells. Data in panels F and G are presented as the mean ± standard errors from two independent experiments. Differences were evaluated using two-way ANOVA. **$p < 0.01$; ***$p < 0.001$; ****$p < 0.0001$.

complex formation with pRIPK1, pRIPK3, and pMLKL, which increased during PSaV replication in a time-dependent manner. Like a characteristic necroptosis feature of viral infections [26, 47, 48, 57], phosphorylated MLKL, the major executioner in the necroptosis pathway, was found to be relocalized at the plasma membrane surface of the PSaV-infected LLC-PK cells. It

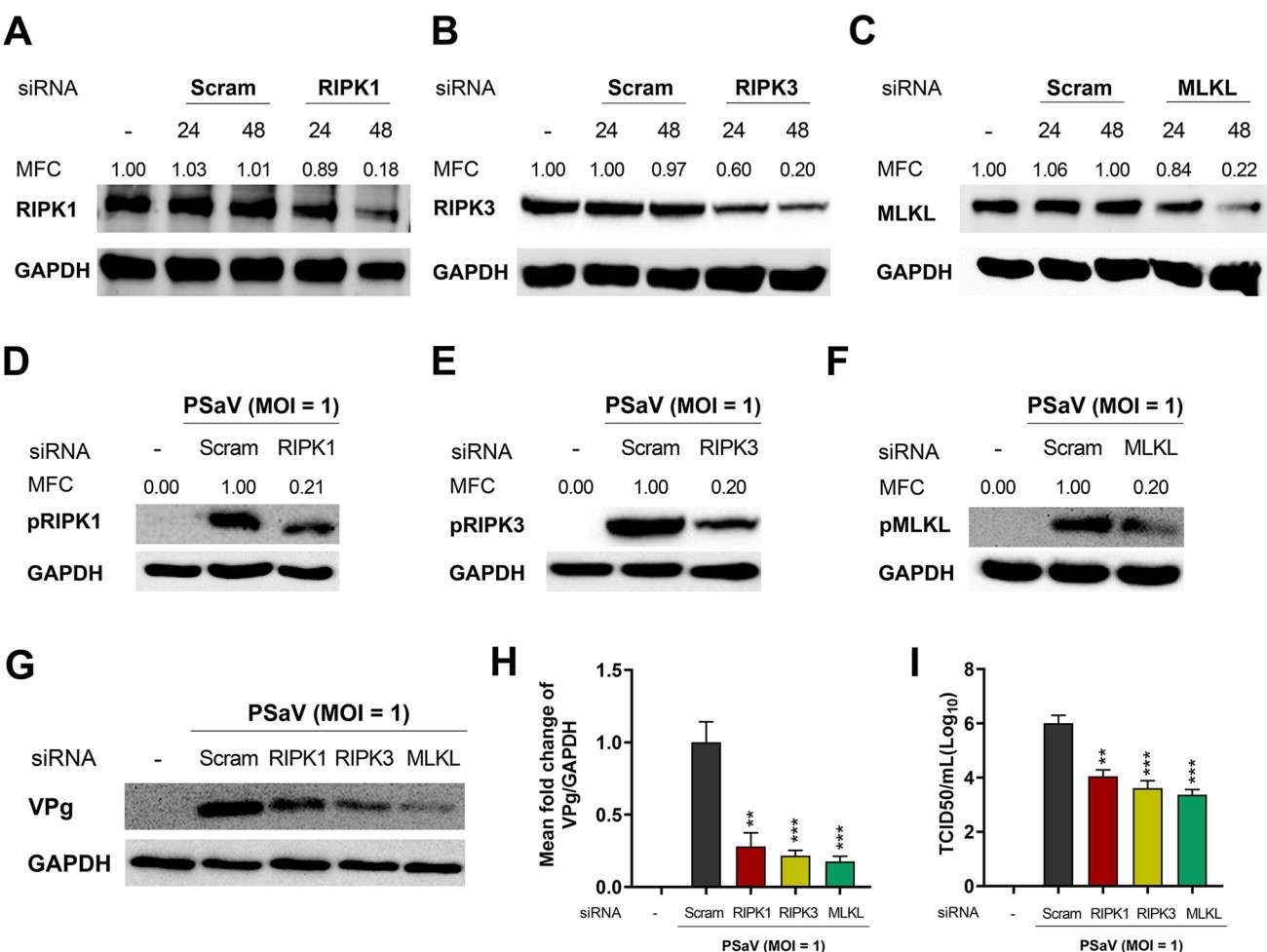

**Fig 7. Silencing of necroptosis molecules in LLC-PK cells inhibits PSaV replication.** (A-C) Mock-infected LLC-PK cells were transfected with either scrambled siRNA or siRNAs against RIPK1 (A), RIPK3 (B), or MLKL (C) and incubated in the presence of 200 μM GCDCA for 24 and 48 h. RIPK1, RIPK3, or MLKL protein levels were evaluated via western blot analysis. GAPDH was used as a loading control. (D-F) The transfected cells with either scrambled siRNA or siRNAs against RIPK1, RIPK3, or MLKL were infected with PSaV Cowden strain (MOI = 1 $TCID_{50}$) and incubated in the presence of 200 μM GCDCA for 24 h. The cell lysates were subjected to western blotting to determine pRIPK1, pRIPK3, and pMLKL, respectively. GAPDH was used as a loading control. (G) PSaV VPg protein levels in the cells described in panels D-F were determined by western blotting. GAPDH was used as a loading control. (H) The relative expression of PSaV VPg (in panel G) was determined via densitometric analysis. (I) The viral titer was determined using $TCID_{50}$ assay. Data are presented as means ± standard errors from two independent experiments. Differences were evaluated using one-way ANOVA. $^{**}p < 0.01$; $^{***}p < 0.001$.

is known that the relocalization of pMLKL into the plasma membrane causes membrane pore formation, resulting in rupture of plasma membrane and eventually leading to morphological characteristics of necroptotic cell death [26]. Taken all together, we demonstrated that PSaV Cowden strain, as a representative of caliciviruses, induced the necroptosis of virus-infected LLC-PK cells through the activation of RIPK1/RIPK3/MLKL signaling pathway.

Both necroptosis and apoptosis in response to viral infections are considered the host defense mechanisms that help control or restrict viral replication [15, 21, 48, 58]. Contrarily, our study showed that interference with each key necroptotic molecule (RIPK1, RIPK3, or MLKL) by specific chemical inhibition or siRNA silencing in PSaV-infected cells reduced the replication of PSaV. These results indicate that necroptosis is proviral for PSaV replication. Bursting of oncolytic virus-infected cells through translocation to and lysis of plasma membrane by the executioner pMLKL in the necroptosis pathway causes the release of not only

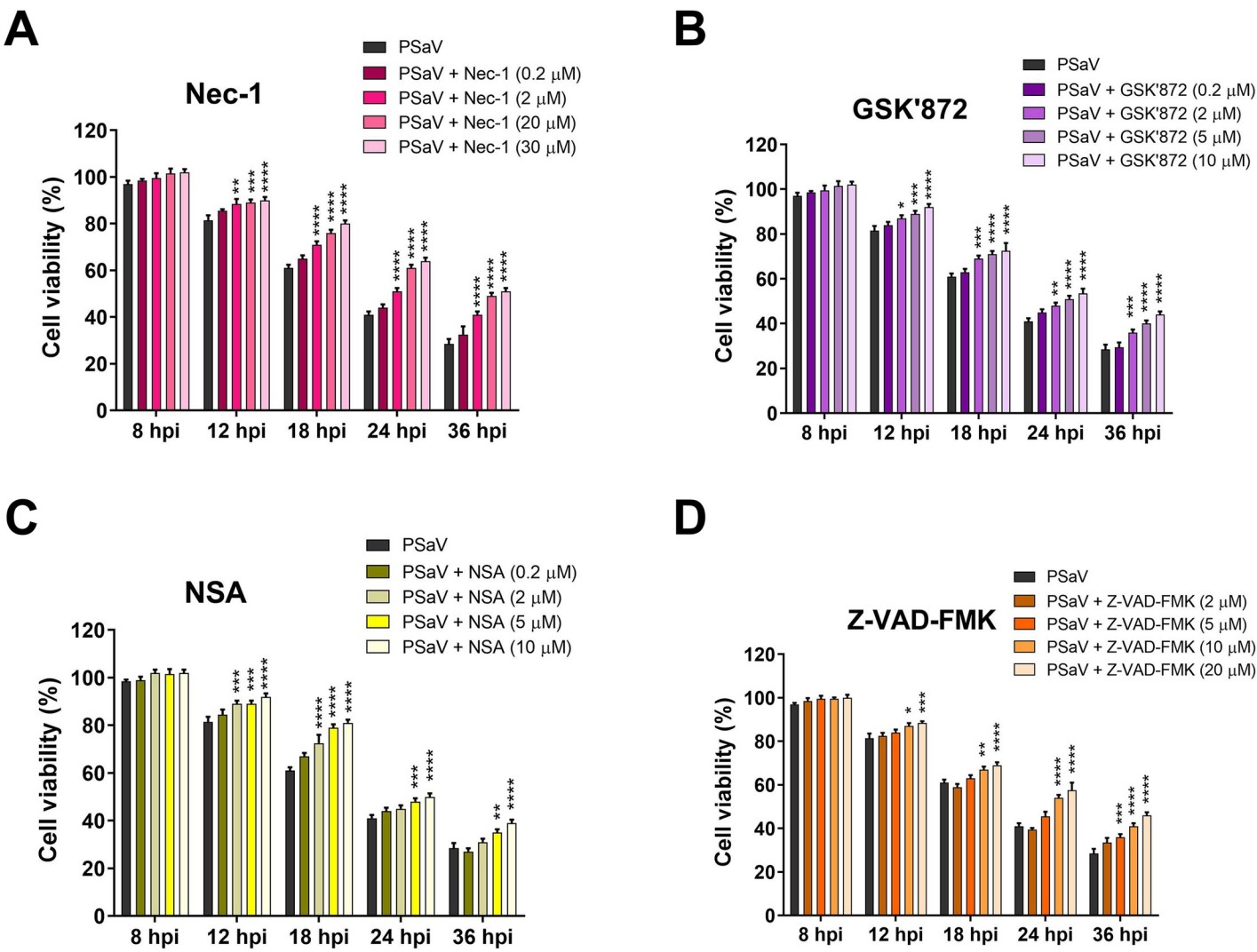

**Fig 8. Inhibition of necroptosis or apoptosis reduces PSaV-induced cell death.** (A-D) LLC-PK cells infected with PSaV Cowden strain (MOI = 1 TCID$_{50}$) were mock-treated or treated with different concentrations of Nec-1 (A), GSK'872 (B), NSA (C) or Z-VAD-FMK (D) and then incubated in the presence of 200 μM GCDCA for the indicated time points. Cell viability determined by MTT assay was compared with that in the mock-treated and virus-infected cells. All experiments were performed in triplicate and the data represent the means ± standard errors of the mean. Differences were evaluated using two-way ANOVA. $^*p < 0.05$; $^{**}p < 0.01$; $^{***}p < 0.001$; $^{****}p < 0.0001$.

intracellular components known as damage-associated molecular patterns (DAMPs) but also viral components and progeny viruses, thereby enabling progeny viruses to spread to other cells or organs [23, 47, 59, 60]. Likely, the explosion of PSaV-infected cells through cell membrane lysis by translocation of pMLKL to the plasma membrane in RIPK1-dependent necroptosis could render the release of PSaV progeny particles to infect nearby cells. Since DAMPs and viral components such as viral nucleic acids, proteins, and even particles released from necroptotic cells are well known to evoke the inflammatory reactions in infected hosts through recognizing by pattern-recognition receptors of innate immune cells, thus resulting in the restriction of progeny spreading to other cells and organs [28, 47, 59, 61]. In contrast to *in vitro* antiviral effect, therefore, whether PSaV-induced necroptosis evokes antiviral innate immune reactions in the virus-infected hosts particularly using RIPK1$^{-/-}$, RIPK3$^{-/-}$, and MLKL$^{-/-}$ knockout animals are required.

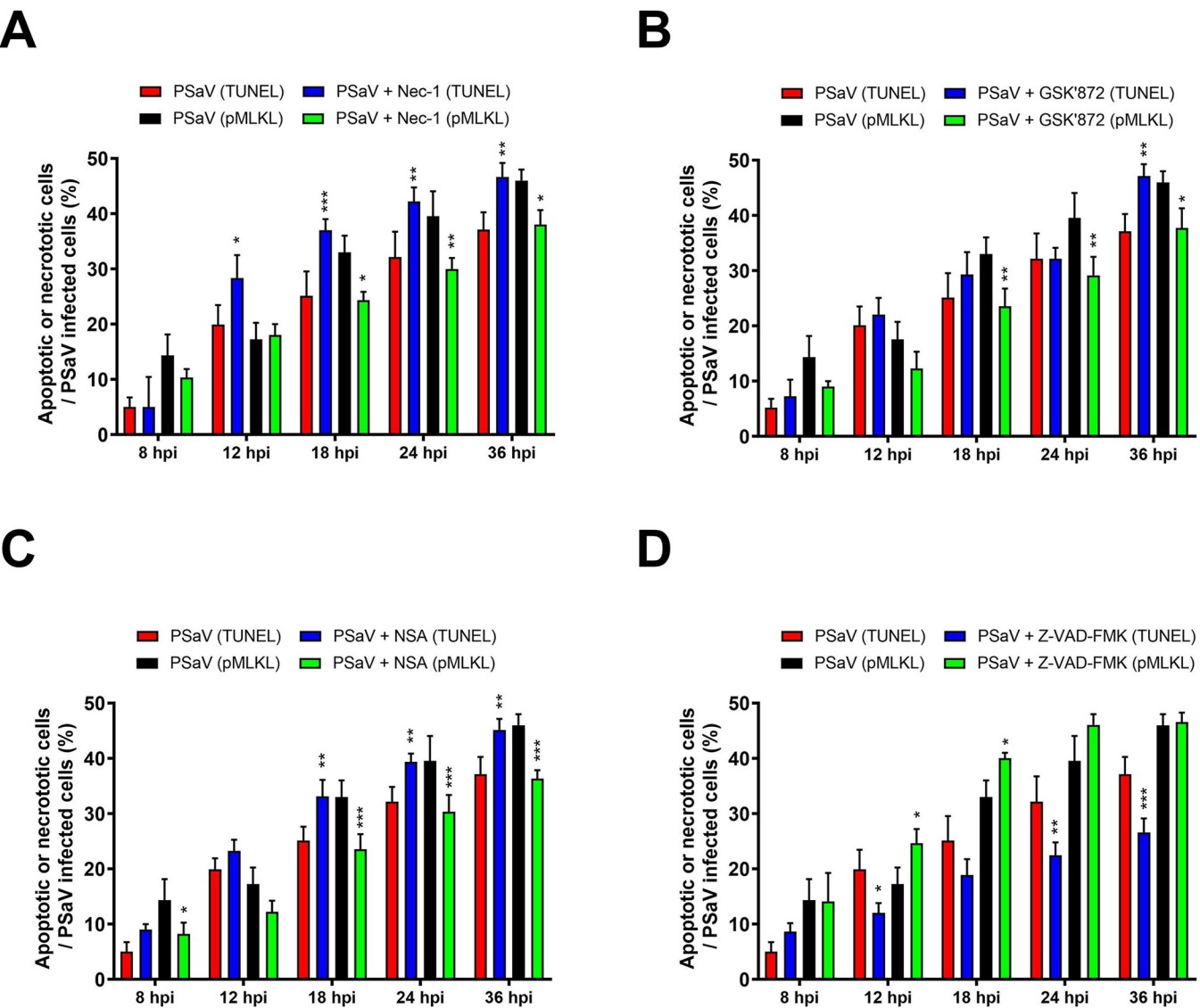

**Fig 9. Inhibition of necroptosis converted PSaV-induced cell death into apoptosis and vice versa.** (A-D) LLC-PK cells were infected with PSaV Cowden strain (MOI = 1 TCID$_{50}$), treated with 30 μM Nec-1 (A), 10 μM GSK'872 (B), 10 μM NSA (C), or 20 μM Z-VAD-FMK (D) or left mock-treated and then harvested at the indicated time points. Quantification of the number of LLC-PK cells positive for both viral VPg antigen and necroptosis molecule pMLKL or for both viral VPg antigen and apoptotic TUNEL signal were analyzed by flow cytometry. The data are shown as the percentage of necrotic or apoptotic LLC-PK cells infected with PSaV in the absence or presence of each chemical. All experiments were performed in triplicate and the data represent the means ± standard errors of the mean. Differences were evaluated using two-way ANOVA. $^{*}p < 0.05$; $^{**}p < 0.01$; $^{***}p < 0.001$.

Like other members of caliciviruses [31–34, 37, 38], PSaV induced apoptosis of virus-infected LLC-PK cells through the activation of caspase 3, which was quantified and visualized by flow cytometry and immunofluorescence assay, respectively. Apoptotic cells are known to confine virus progeny inside the cells, thus restricting the release and spread of intracellular progeny particles [18, 62]. As subsequent apoptotic bodies develop new ligands for binding, phagocytic cells engulf them and prevent the release of infectious virus progeny into the surrounding environment in vivo [18, 62]. In the present study, inhibition of apoptosis using a pan-caspase inhibitor (Z-VAD-FMK) in PSaV-infected cells significantly increased the replication of PSaV. Supporting the general concept of the role of apoptosis in virus

replication, our data suggest that apoptosis also acts as a host defense mechanism against PSaV infection.

Despite the marked morphological and molecular differences between necroptotic and apoptotic cell deaths, apoptosis and necroptosis are known to cross-talk each other in virus-infected cells [47, 53, 55, 63]. It has been previously demonstrated that necroptosis can be diverted to apoptosis when necroptosis signaling was blocked and vice versa [47, 53, 55]. Therefore, we attempted to address the interplay between necroptosis and apoptosis in the PSaV-infected LLC-PK cells using each chemical inhibitor specific for necroptotic and apoptotic molecules. Blockade of each key necroptotic molecule (RIPK1, RIPK3, or MLKL) in PSaV-infected LLC-PK cells reduced the replication of PSaV but increased the number of apoptotic cells and cell viability. This data suggested that the shift from necroptosis to apoptosis in the PSaV-infected LLC-PK cells could hinder the replication and spread of PSaV from apoptotic cells, thereby reducing the spread of the virus to neighboring cells. These results are consistent with a previous report that blockade of each necroptosis signaling molecule by specific chemical inhibition or siRNA silencing in rotavirus-infected cells resulted in increased apoptotic cell death and thereby decreased virus yields [47]. Moreover, it was reported that treatment of HIV-1 infected cells with Nec-1, an inhibitor of RIPK1, triggered a switch of necroptosis to apoptosis while inhibiting syncytia formation [53]. In contrast, blockade of apoptosis in PSaV-infected LLC-PK cells using a pan-caspase inhibitor (Z-VAD-FMK) increased both PSaV titers and necroptotic cells. As cell-bursting pMLKL was found to be translocated to the cell membrane, the shift from apoptosis to necroptosis in the PSaV-infected LLC-PK cells could enable to release of PSaV progeny from the necroptotic cells, thereby facilitating the spread of the progeny to neighboring cells. This is consistent with reports that inhibition of apoptosis in TMEV-, HIV-1-, and rotavirus-infected cells significantly increases viral yield by switching the cell death from apoptosis to necroptosis [47, 53, 55]. Recently, extensive crosstalk between apoptosis, necroptosis, and pyroptosis, termed PANoptosis, has been identified in the cells infected with various viruses [63]. The best example is the sensing of influenza A virus Z-RNA by ZBP1 allows the activation of proteins involved in necroptosis, apoptosis, and pyroptosis to form the ZBP1-PANoptosome and mediate PANoptosis. Likely, PSaV infection could simultaneously activate the proteins involved in necroptosis, apoptosis, and pyroptosis, which allows forming the PSaV-PANoptosome and mediates PANoptosis [63]. In addition, calicivirus proteins such as RHDV VP2 and NS6, feline calicivirus leader of capsid, murine norovirus ORF1 polyprotein, and human norovirus NTPase are known to induce the intrinsic apoptosis [40, 64–67]. Moreover, the same viral protein, i.e. rotavirus NSP4, is known to evoke both apoptosis and necroptosis [47]. Currently, therefore, whether PSaV Z-RNA could induce ZBP1-PANoptosome and which viral and host molecules could determine PSaV-PANoptosis forms the basis of our ongoing work.

In conclusion, our study reveals that the PSaV Cowden strain as a representative of caliciviruses induces necroptosis of virus-infected LLC-PK cells through a RIPK1-dependent pathway, which acts as proviral compared to the antiviral function of PSaV-induced apoptosis. Future studies on how the PSaV and other caliciviruses control host cell modalities will facilitate the development of therapeutic agents for the benefit of the host immune system to limit virus replication and clear infected cells.

## Supporting information

**S1 Data. Minimal data.**
(ZIP)

**S1 Raw images.**
(PDF)

## Author Contributions

**Investigation:** Thu Ha Nguyen.

**Supervision:** Mahmoud Soliman, Kyoung-Oh Cho.

**Writing – original draft:** Muhammad Sharif, Yeong-Bin Baek.

**Writing – review & editing:** Mahmoud Soliman.

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
