## [Decision Letter · Decision Letter 0]

12 Oct 2022

PONE-D-22-23172Porcine sapovirus-induced RIPK1-dependent necroptosis is proviral in LLC-PK cellsPLOS ONE

Dear Dr. Cho,

Thank you for submitting your manuscript to PLOS ONE. After careful consideration, we feel that it has merit but does not fully meet PLOS ONE’s publication criteria as it currently stands. Therefore, we invite you to submit a revised version of the manuscript that addresses the points raised during the review process.

We look forward to receiving your revised manuscript.

Kind regards,

Saeid Ghavami, PhD

Academic Editor

PLOS ONE

Journal Requirements:

https://pubmed.ncbi.nlm.nih.gov/34668777/?

In your revision ensure you cite all your sources (including your own works), and quote or rephrase any duplicated text outside the methods section. Further consideration is dependent on these concerns being addressed.

"No authors have competing interests."

Additional Editor Comments:

Dear Professor Kyoung-Oh,

Thank you for submission to PLOS ONE and I sincerely do apologize for late processing of your article. It was very hard to secure good reviewer and get feedback and comments. Besides addressing the respected reviewers,

1- Please use Necrostatin and address its effect in your model.

2- Please provide original WB without any crop.

Thank you

Saeid Ghavami, PhD, AE

Reviewers' comments:

Reviewer's Responses to Questions

**Comments to the Author**

1. Is the manuscript technically sound, and do the data support the conclusions?

Reviewer #1: Yes

Reviewer #2: Partly

2. Has the statistical analysis been performed appropriately and rigorously? 

Reviewer #1: No

Reviewer #2: Yes

3. Have the authors made all data underlying the findings in their manuscript fully available?

Reviewer #1: Yes

Reviewer #2: Yes

4. Is the manuscript presented in an intelligible fashion and written in standard English?

Reviewer #1: Yes

Reviewer #2: Yes

5. Review Comments to the Author

Reviewer #1: The present manuscript ‘Porcine sapovirus-induced RIPK1-dependent necroptosis is proviral in LLC-PK cells’ describes an interesting work and has potential to become valuable reading materials. However, there are several minor concerns and confused questions before it can be published by PLOS ONE.

Minor concerns:

1. The P values for all graph statistics are incorrect and the number of ‘*’ on the picture is also very weird, newly statistical analysis is recommended.

2. The full and abbreviated names of proper nouns are confusing. It is recommended that the full name only appear once.

3. Immunoprecipitation assay: line 237-239, check the writing of ‘°C’.

4. Page 37, line 653, replace “TCID50” with “TCID50”

5. Please check for grammar errors, for example line 720 and line 721.

6. Figure 6E and 6F, please explain why the similar bands in the Z-VAD-FMK treatment group in Fig. 6E have a statistically significant darkening trend in Fig. 6F.

7. Figure 8D, PSaV+ Z-VAD-FMK treatment group, please explain why the higher the concentration of Z-VAD-FMK, the better the cell viability, why both anti- necroptosis and pro-necroptosis can improve cell activity, isn't this contrary to your conclusion in this article?

8. Abstract: “Interfering of PSaV-infected cells with each necroptotic molecule (RIPK1, RIPK3, or MLKL) by treatment with each specific chemical inhibitor or knockdown with each specific siRNA significantly reduced replication of PSaV but increased apoptosis and cell viability, implying proviral action of PSaV-induced necroptosis. In contrast, treatment of PSaV-infected cells with pan-caspase inhibitor Z-VAD-FMK increased PSaV replication and necroptosis, indicating antiviral action of PSaV-induced apoptosis.” Your data may demonstrate a proviral effect of necroptosis, but no direct data suggest an antiviral effect of apoptosis, please explain more directly how you came to this conclusion.

Reviewer #2: Title: Porcine sapovirus-induced RIPK1- dependent necroptosis is proviral in LLC-PK cells

Author: Sharif et al

The manuscript shows that Porcine Sapovirus (PSaV) induces necroptosis and apoptosis in the Porcine Kidney Cells (LLC-PK). Inhibition of critical regulatory and executioner proteins of necroptosis with their respective chemical inhibitors (Nec-1, GSK'872, NSA) or silencing those three proteins (RIPK1, RIPK3, and MLKL) with siRNA prevented the phosphorylation of the proteins and virus replication and targeted viral proteins. Inhibition of necroptosis during virus infection increases cell viability relative to the virus-treated cell alone, and inhibition of apoptosis increases necroptosis and promotes virus replication. The authors propose a counterbalance between apoptosis and necroptosis in virus-infected cells.

Major

1) Fig 1, 2, 3, 8, and 9. The data shows that virus infection kills the cells irrespective of whether both necroptosis and apoptosis are inhibited, and remanding cells seem to exhibit both necroptosis and apoptosis characteristics. Please comment on this phenomenon.

2) Figure 6F and G. On page 15, line 314, it reads, "Interestingly, inhibition of necroptosis by the treatment with each chemical necroptosis inhibitor suppressed the expression of PSaV VPg protein (Fig6E and 6F) and PSaV titer (Fig 6G)." Please explain why there is a difference in the protein level produced by the virus (VPg in Figure 6F) but not necessarily in the amount of virus replication measured by the TCID50 (Fig6G).

3) line 316: "However, treatment with chemical apoptosis inhibitor reduced viral protein expression and titer (Fig 6E-G)." The caspase inhibitor increased the viral protein at 24hrs and the PSaV titer at 36 hours relative to the PSaV treatment only in those time points. Please clarify and discuss the significance.

4) Fig 6F and Fig 6G as in other figures? Please show the mock control for these experiments. What is the expected cell viability for the LLC-PK cell not treated with the PSaV?

5) Necrosis vs. Necroptosis.

On page 16, line 326, the authors wrote, "A shift in cell death modality from apoptosis to necrosis and vice versa has been observed in response to human immunodeficiency virus type-1 (HIV-1), Theiler's murine encephalomyelitis virus (TMEV), and rotavirus infections in the presence of chemicals against the necroptosis or apoptosis." Necrosis is an unregulated, uncontrolled, and irreversible cell death process. There is swelling of the organelles, plasma membrane rupture, and eventual lysis of the cell. Necrosis is triggered in the context of excessive external stress, such as heat, ischemia, and pathogen infection. It doesn't need the Tumor Necrosis Factor Receptor activation or a specific cell death complex to trigger it. Zhou W, Yuan J. Semin Necroptosis in health and diseases. Cell Dev Biol. 2014 Nov;35:14-23. Festjens N, Vanden Berghe T, Vandenabeele P. Necrosis, a well-orchestrated form of cell demise: signaling cascades, important mediators and concomitant immune response. Biochim Biophys Acta. 2006 Sep-Oct;1757(9-10):1371-87. In contrast, "necroptosis is a more physiological and programmed type of necroptotic death and shares several key processes with apoptosis. It occurs due to activating the kinase domain of the receptor-interacting protein 1 (RIP1) and the assembly of the RIP1/RIP3-containing signaling complex. It is triggered by tumor necrosis factor (TNF) family members, needs caspase eight inhibition, and assembly of necrosome (RIPK1-RIPK3 complex IIb) with activation of MLKL." Elmore SA, Dixon D, Hailey JR, Harada T, Herbert RA, Maronpot RR, Nolte T, Rehg JE, Rittinghausen S, Rosol TJ, Satoh H, Vidal JD, Willard-Mack CL, Creasy DM. Recommendations from the INHAND Apoptosis/Necrosis Working Group. Toxicol Pathol. 2016 Feb;44(2):173-88. Even though necroptosis is classified as a type of necrosis, it would be better to use the term necroptosis in that sentence because necrosis cannot be shifted to apoptosis, and apoptosis cannot be shifted to necrosis. Even the four citations at the end of the sentence (47, 53, 55, 56) have necroptosis in their tittles to convey the message that necroptosis is the process that can be shifted to apoptosis.

6-) In figure 8, it could be appreciated that the cells treated with PSaV die anyways, irrespective of the inhibitor used to prevent apoptosis of necroptosis. Therefore, contrary to the line on page 16, line 334, which reads, "Interestingly, necroptosis inhibition of PSaV-infected cells by the individual treatment with the inhibitor against the RIPK1, RIPK3, or MLKL showed an increase in cell viability (Fig 8A-C) and the proportion of apoptotic cells." I suggest describing the findings in Figure 8, stating that inhibiting apoptosis or necroptosis did not prevent cell death by PSaV but lessened its impact. It's evident that by 36hrs, all of the treatments in Fig 8 show cell viability at less than 50%.

7-) In Figure 9, the same observation can be made for Fig 8. Even when it is true that inhibiting necroptosis and apoptosis changes one cell death process into another, the overall trend is that both processes are increased and that, irrespective of the treatment, the cells will show increased necroptosis or necroptosis. A better way of describing the findings in Figure 9 is to say that inhibiting necroptosis or apoptosis changes the proportion of cells passing through apoptosis or necroptosis.

Minor

1) In Figures 1C, 2C, 6F, 6G, 8ABCD, and 9ABCD. Did the author try using two-way ANOVA? These figures show more than two independent categorical variables.

6. PLOS authors have the option to publish the peer review history of their article (what does this mean?). If published, this will include your full peer review and any attached files.

Reviewer #1: No

Reviewer #2: No

---

## [Author Response · Author response to Decision Letter 0]

26 Nov 2022

Responses to Reviewers’ Comments

MS title: Porcine sapovirus-induced RIPK1-dependent necroptosis is proviral in LLC-PK cells

MS No.: PONE-D-22-23172

In the revised manuscript, the modified words or sentences in response to reviewers’ suggestions and corrections are indicated in red (first reviewer) and blue (second reviewer).

Journal Requirements:

Response:

• Thank you for pointing this out. 

• We have modified the revised manuscript to meet PLOS ONE's style requirements (including the symbol legend, indicating affiliations by number, the font size of the heading for all major sections and subheadings, and insertion of each figure caption directly after the paragraph in which they are first cited). 

https://pubmed.ncbi.nlm.nih.gov/34668777/?

In your revision ensure you cite all your sources (including your own works), and quote or rephrase any duplicated text outside the methods section. Further consideration is dependent on these concerns being addressed.

Response:

• Thank you for pointing this out. 

• We have ensured to cite all the sources and rephrased the duplicated text (Lines 158, 188, 211, 261, and 361).

"No authors have competing interests."

Response:

• Thank you very much. We have included this information in the cover letter.

Response:

• Thank you very much. We have updated our Data Availability statement in the cover letter.

5. PLOS ONE now requires that authors provide the original uncropped and unadjusted images underlying all blot or gel results reported in a submission’s figures or Supporting Information files. This policy and the journal’s other requirements for blot/gel reporting and figure preparation are described in detail at https://journals.plos.org/plosone/s/figures#loc-blot-and-gel-reporting-requirements and https://journals.plos.org/plosone/s/figures#loc-preparing-figures-from-image-files. 

When you submit your revised manuscript, please ensure that your figures adhere fully to these guidelines and provide the original underlying images for all blot or gel data reported in your submission. See the following link for instructions on providing the original image data: https://journals.plos.org/plosone/s/figures#loc-original-images-for-blots-and-gels. 

Response:

• Thank you for pointing this out. We have provided the original uncropped images in the Supporting Information file.

Additional Editor Comments:

1- Please use Necrostatin and address its effect in your model.

Response:

• Thank you for your suggestion. We have used necrostatin in the original manuscript (Figs. 5, 6, 8, and 9)

2- Please provide original WB without any crop.

Response:

• Thank you for pointing this out. We have provided the original uncropped images in the Supporting Information file.

 

5. Review Comments to the Author

Reviewer #1: 

The present manuscript ‘Porcine sapovirus-induced RIPK1-dependent necroptosis is proviral in LLC-PK cells’ describes an interesting work and has potential to become valuable reading materials. However, there are several minor concerns and confused questions before it can be published by PLOS ONE.

Minor concerns:

1. The P values for all graph statistics are incorrect and the number of ‘*’ on the picture is also very weird, newly statistical analysis is recommended.

Response: 

• Thank you for pointing this out and we apologize for this mistake.

• We have repeated the statistical analysis and recalculated P values in the revised figures.

2. The full and abbreviated names of proper nouns are confusing. It is recommended that the full name only appear once.

Response:

• Thank you for pointing this out. We have spelled it out in the revised manuscript (lines 20, 100, 112, 135, 186, 187, 223, 288, 296, 327, 363, 380, 389, 396, 412, 432, 525, and 526). 

3. Immunoprecipitation assay: line 237-239, check the writing of ‘°C’.

Response: 

• Thank you for pointing this out. We have corrected it in the revised manuscript (lines 238-240).

4. Page 37, line 653, replace “TCID50” with “TCID50”

Response: 

• Thank you for pointing this out. We have replaced it in the revised manuscript (line 284).

5. Please check for grammar errors, for example line 720 and line 721.

Response: 

• Thank you for pointing this out. We have corrected them in the revised manuscript (lines 24, 25, 31, 49, 60, 85, 91, 133, 248, 261, 302, 331, 397, 398, 429, 456, 469, 487, 499, 510).

6. Figure 6E and 6F, please explain why the similar bands in the Z-VAD-FMK treatment group in Fig. 6E have a statistically significant darkening trend in Fig. 6F.

Response: 

• Thank you for pointing this out. 

• In this study, we measured mean fold changes of target proteins as following formula; Fold change = [(intensity of target band/intensity of housekeeping protein (GAPDH)] ÷ [(intensity of control band/intensity of housekeeping protein (GAPDH)] Using fold change in each replicate measurement, mean fold change is calculated as follows, mean fold change = sum of replicate values for an experimental sample ÷ number of replicates for that sample.

• We have changed it with another replicate.

7. Figure 8D, PSaV+ Z-VAD-FMK treatment group, please explain why the higher the concentration of Z-VAD-FMK, the better the cell viability, why both anti- necroptosis and pro-necroptosis can improve cell activity, isn't this contrary to your conclusion in this article?

Response: 

• Thank you for pointing this out. 

• The higher concentration of Z-VAD-FMK (20 µM) is the optimal non-cytotoxic concentration to inhibit apoptosis and hence the better cell viability.

• As a characteristic of virus-induced apoptosis, apoptotic cell death eventually prevents release of intracellular virus particles. Apoptosis is an imperative mechanism involved in eliminating virus-infected cells from the host. Therefore, treatment of the optimal non-cytotoxic concentration of Z-VAD-FMK (20 µM) inhibits the apoptotic cell death in PSaV-infected cells, partially improving cell viability. 

• The treatment of anti-necroptosis chemicals inhibits the necroptotic cell death in PSaV-infected, blocks the release and spread of progeny viruses between cells, and hence partially improves the cell viability.

8. Abstract: “Interfering of PSaV-infected cells with each necroptotic molecule (RIPK1, RIPK3, or MLKL) by treatment with each specific chemical inhibitor or knockdown with each specific siRNA significantly reduced replication of PSaV but increased apoptosis and cell viability, implying proviral action of PSaV-induced necroptosis. In contrast, treatment of PSaV-infected cells with pan-caspase inhibitor Z-VAD-FMK increased PSaV replication and necroptosis, indicating antiviral action of PSaV-induced apoptosis.” Your data may demonstrate a proviral effect of necroptosis, but no direct data suggest an antiviral effect of apoptosis, please explain more directly how you came to this conclusion.

Response: 

• Thank you for pointing this out. 

• In Fig. 6E-G, we have shown that the PSaV viral protein expression and viral titer were increased in LLC-PK cells treated with the pan-caspase inhibitor Z-VAD-FMK. However, further experiments need to be performed in the near future to address this point.

 

Reviewer #2:

Title: Porcine sapovirus-induced RIPK1- dependent necroptosis is proviral in LLC-PK cells. 

Author: Sharif et al. 

The manuscript shows that Porcine Sapovirus (PSaV) induces necroptosis and apoptosis in the Porcine Kidney Cells (LLC-PK). Inhibition of critical regulatory and executioner proteins of necroptosis with their respective chemical inhibitors (Nec-1, GSK'872, NSA) or silencing those three proteins (RIPK1, RIPK3, and MLKL) with siRNA prevented the phosphorylation of the proteins and virus replication and targeted viral proteins. Inhibition of necroptosis during virus infection increases cell viability relative to the virus-treated cell alone, and inhibition of apoptosis increases necroptosis and promotes virus replication. The authors propose a counterbalance between apoptosis and necroptosis in virus-infected cells.

Major

1) Fig 1, 2, 3, 8, and 9. The data shows that virus infection kills the cells irrespective of whether both necroptosis and apoptosis are inhibited, and remanding cells seem to exhibit both necroptosis and apoptosis characteristics. Please comment on this phenomenon.

Response: 

• Thank you for pointing this out. 

• Cell death is a naturally occurring phenomenon in multicellular organisms. Cells die as a result of internal and external stimuli. The most common type of programmed cell death is apoptosis. It is activated by a variety of physical, chemical, and biological factors, and pathogens such as bacteria and viruses, and its cellular response is tightly controlled. Caspases, a family of proteases activated during apoptosis, regulate the controlled degradation of cellular components that occurs during apoptosis. Significant cell morphological changes occur as a result of apoptosis. A cell undergoing apoptosis loses cell contacts and changes shape in the early stages. In the nucleus, chromatin condenses and moves toward the nuclear envelope. DNA degradation begins with nucleus condensation known as pyknosis (Galluzzi L, Vitale I, et al 2018 DOI: 10.1038/s41418-017-0012-4. and Gergely Imre, et al 2020, https://doi.org/10.1016/j.cellsig.2020.109772.)

• Necroptosis is a type of cell injury that is defined as controlled cell death caused by internal or external stresses such as mechanical injuries, chemical agents, or pathogens. During necroptosis, the loss of plasma membrane integrity induces cellular contents to escape to the extracellular space, causing inflammatory responses. Cell disintegration is preceded by a series of morphological changes, including disruption of cell organelles, such as swelling of the ER and mitochondria, or decay of the Golgi apparatus (Galluzzi L, Vitale I, et al 2018 DOI: 10.1038/s41418-017-0012-4. and Gergely Imre, et al 2020, https://doi.org/10.1016/j.cellsig.2020.109772.). 

2) Figure 6F and G. On page 15, line 314, it reads, "Interestingly, inhibition of necroptosis by the treatment with each chemical necroptosis inhibitor suppressed the expression of PSaV VPg protein (Fig6E and 6F) and PSaV titer (Fig 6G)." Please explain why there is a difference in the protein level produced by the virus (VPg in Figure 6F) but not necessarily in the amount of virus replication measured by the TCID50 (Fig6G).

Response: 

• Thank you for pointing this out. 

• Figure 6 E shows that blocking each key necroptotic molecule (RIPK1, RIPK3, or MLKL) in PSaV-infected LLC-PK cells reduced PSaV (Vpg) replication in a time-dependent manner compared to PSaV alone infected cells because these (RIPK1, RIPK3, and MLKL) signaling pathways induce necroptosis and help in virus replication. These results are similar to what we have been reported for cells infected with rotavirus [Opposite Effects of Apoptotic and Necroptotic Cellular Pathways on Rotavirus Replication (Soliman et al., 2022; https://doi.org/10.1128/jvi.01222-21)].

• However, the protein level is different from TCID50, because western blotting is an analytical technique that is used to separate and identify proteins from a mixture or pellet, and the number of infectious virus particles is frequently quantified by using the Median Tissue Culture Infectious Dose (TCID50) assay. 

3) line 316: "However, treatment with chemical apoptosis inhibitor reduced viral protein expression and titer (Fig 6E-G)." The caspase inhibitor increased the viral protein at 24hrs and the PSaV titer at 36 hours relative to the PSaV treatment only in those time points. Please clarify and discuss the significance.

Response: 

• Thank you for pointing this out. 

• Accordingly, we have repeated the caspase inhibitor (Z-VAD-FMK) experiment in Fig. 6 E which shows increased viral protein time-dependently from 12 to 36 hpi. These results are similar to what we have reported for cells infected with rotavirus [Opposite Effects of Apoptotic and Necroptotic Cellular Pathways on Rotavirus Replication (Soliman et al., 2022; https://doi.org/10.1128/jvi.01222-21)].

• Virus titter is generally determined using TCID50 assay, in this assay usually we checked the cytopathic effect percentage, which means progeny virus increased with increased cytopathic effect at 36 hpi so in our data virus replication or progeny virus increased time-dependently. We have reported in our previous article, viral protein and titter increased with time (Alfajaro et al., 2019, https://doi.org/10.1128/JVI.01773-18).

4) Fig 6F and Fig 6G as in other figures? Please show the mock control for these experiments. What is the expected cell viability for the LLC-PK cell not treated with the PSaV?

Response: 

• Thank you for pointing this out.

• Fig. 6F is a graphical representation of Fig. 6E, whereas Fig. 6G is a virus titer measured by TCID50 assay. 

• We have modified the mock control in the revised Fig. 6F and G.

• The cell viability for LLC-PK cells mock-infected has been shown in Fig. 1A.

5) Necrosis vs. Necroptosis.

On page 16, line 326, the authors wrote, "A shift in cell death modality from apoptosis to necrosis and vice versa has been observed in response to human immunodeficiency virus type-1 (HIV-1), Theiler's murine encephalomyelitis virus (TMEV), and rotavirus infections in the presence of chemicals against the necroptosis or apoptosis." Necrosis is an unregulated, uncontrolled, and irreversible cell death process. There is swelling of the organelles, plasma membrane rupture, and eventual lysis of the cell. Necrosis is triggered in the context of excessive external stress, such as heat, ischemia, and pathogen infection. It doesn't need the Tumor Necrosis Factor Receptor activation or a specific cell death complex to trigger it. Zhou W, Yuan J. Semin Necroptosis in health and diseases. Cell Dev Biol. 2014 Nov;35:14-23. Festjens N, Vanden Berghe T, Vandenabeele P. Necrosis, a well-orchestrated form of cell demise: signaling cascades, important mediators and concomitant immune response. Biochim Biophys Acta. 2006 Sep-Oct;1757(9-10):1371-87. In contrast, "necroptosis is a more physiological and programmed type of necroptotic death and shares several key processes with apoptosis. It occurs due to activating the kinase domain of the receptor-interacting protein 1 (RIP1) and the assembly of the RIP1/RIP3-containing signaling complex. It is triggered by tumor necrosis factor (TNF) family members, needs caspase eight inhibition, and assembly of necrosome (RIPK1-RIPK3 complex IIb) with activation of MLKL." Elmore SA, Dixon D, Hailey JR, Harada T, Herbert RA, Maronpot RR, Nolte T, Rehg JE, Rittinghausen S, Rosol TJ, Satoh H, Vidal JD, Willard-Mack CL, Creasy DM. Recommendations from the INHAND Apoptosis/Necrosis Working Group. Toxicol Pathol. 2016 Feb;44(2):173-88. Even though necroptosis is classified as a type of necrosis, it would be better to use the term necroptosis in that sentence because necrosis cannot be shifted to apoptosis, and apoptosis cannot be shifted to necrosis. Even the four citations at the end of the sentence (47, 53, 55, 56) have necroptosis in their tittles to convey the message that necroptosis is the process that can be shifted to apoptosis.

Response: 

• Thank you for pointing this out and we apologize for this mistake.

• We have modified it in the revised manuscript (line 417).

6-) In figure 8, it could be appreciated that the cells treated with PSaV die anyways, irrespective of the inhibitor used to prevent apoptosis of necroptosis. Therefore, contrary to the line on page 16, line 334, which reads, "Interestingly, necroptosis inhibition of PSaV-infected cells by the individual treatment with the inhibitor against the RIPK1, RIPK3, or MLKL showed an increase in cell viability (Fig 8A-C) and the proportion of apoptotic cells." I suggest describing the findings in Figure 8, stating that inhibiting apoptosis or necroptosis did not prevent cell death by PSaV but lessened its impact. It's evident that by 36hrs, all of the treatments in Fig 8 show cell viability at less than 50%.

Response: 

• Thank you for pointing this out. We have modified it as suggested in the revised manuscript (lines 425-427).

7-) In Figure 9, the same observation can be made for Fig 8. Even when it is true that inhibiting necroptosis and apoptosis changes one cell death process into another, the overall trend is that both processes are increased and that, irrespective of the treatment, the cells will show increased necroptosis or necroptosis. A better way of describing the findings in Figure 9 is to say that inhibiting necroptosis or apoptosis changes the proportion of cells passing through apoptosis or necroptosis.

Response: 

• Thank you for pointing this out. We have modified it as suggested in the revised manuscript (lines 425-427).

Minor

1) In Figures 1C, 2C, 6F, 6G, 8ABCD, and 9ABCD. Did the author try using two-way ANOVA? These figures show more than two independent categorical variables.

Response: 

• Thank you for pointing this out.

• We have repeated the statistical analysis and recalculated P values in the revised figures.

---

## [Decision Letter · Decision Letter 1]

15 Dec 2022

Porcine sapovirus-induced RIPK1-dependent necroptosis is proviral in LLC-PK cells

PONE-D-22-23172R1

Dear Dr. Cho,

We’re pleased to inform you that your manuscript has been judged scientifically suitable for publication and will be formally accepted for publication once it meets all outstanding technical requirements.

Kind regards,

Saeid Ghavami, PhD

Academic Editor

PLOS ONE

Additional Editor Comments (optional):

Reviewers' comments:

Reviewer's Responses to Questions

**Comments to the Author**

1. If the authors have adequately addressed your comments raised in a previous round of review and you feel that this manuscript is now acceptable for publication, you may indicate that here to bypass the “Comments to the Author” section, enter your conflict of interest statement in the “Confidential to Editor” section, and submit your "Accept" recommendation.

Reviewer #1: All comments have been addressed

Reviewer #3: All comments have been addressed

2. Is the manuscript technically sound, and do the data support the conclusions?

Reviewer #1: Yes

Reviewer #3: Yes

3. Has the statistical analysis been performed appropriately and rigorously? 

Reviewer #1: Yes

Reviewer #3: Yes

4. Have the authors made all data underlying the findings in their manuscript fully available?

Reviewer #1: Yes

Reviewer #3: Yes

5. Is the manuscript presented in an intelligible fashion and written in standard English?

Reviewer #1: Yes

Reviewer #3: Yes

6. Review Comments to the Author

Reviewer #1: (No Response)

Reviewer #3: The revision significantly improves the paper. The authors have carefully addressed all the comments in a proper way.

7. PLOS authors have the option to publish the peer review history of their article (what does this mean?). If published, this will include your full peer review and any attached files.

Reviewer #1: No

Reviewer #3: **Yes: **Saeid Ghavami
